

# Analytical treatment of proton double-quantum NMR intensity build-up: multi-spin couplings and the flip-flop term

Nail Fatkullin[1], Ivan Brekotkin[2], Kay Saalwächter[1]

[1]Institut für Physik–NMR, Martin-Luther-Universität Halle-Wittenberg,  06120 Halle (Saale),
[2]Institute of Physics, Kazan Federal University, Kazan, Tatarstan 420008, Russia

*Correspondence to*: Nail Fatkullin (nfatkull@gmail.com),

**Abstract.**

A modified Anderson-Weiss approximation for describing double quantum (DQ) NMR experiments in systems with many $I = 1/2$ spins is proposed, taking inter-spin flip-flop processes into special consideration. In this way, an analytical result is derived for multi-spin systems for the first time. It is shown that in the initial stages of DQ intensity build-up, the probability of flip-flop processes in DQ experiments is half as large as in analogous Hahn-echo or free-induction-decay experiments. Their influence on the experimentally observed DQ NMR signal becomes dominant at times $t > \left(9/2\right)^{1/2} T_2^{eff} \approx 2.12 T_2^{eff}$ , where $T_2^{eff}$ is the effective spin-spin relaxation time measured by the Hahn echo. Calculations and a comparison with spin-dynamics simulations of small spin systems up to 8 spins reveal a satisfactory agreement.

## 1 Introduction.

The seminal papers by Baum and Pines (Baum et al., 1985; Baum and Pines, 1986) started the field of Double-Quantum (DQ), or more generally Multiple-Quantum (MQ). At the qualitative level, the idea of the method is quite simple. The system of spins under study is continuously irradiated by special sequences of radiofrequency (RF) pulses, which allows one to change the relative importance of different parts of the initial Hamiltonian of spin-lattice interactions, inducing different quantum transitions in the spin system. In other words, irradiation creates a new effective interaction Hamiltonian that induces more selective quantum transitions in the spin system than the original one. In the mentioned initial papers (Baum et al., 1985; Baum and Pines, 1986) the method was mathematically justified for solids, in which thermal motions of spins can be neglected in comparison with the initial spatial distances between them. An additional feature of solids is the large difference between the spin-spin and spin-lattice relaxation times $T_2^{eff} \ll T_1$. This allows us, at times much shorter than the



spin-lattice relaxation time $t \ll T_1$, to neglect the influence of the so-called non-secular part of the spin-lattice interaction

Hamiltonian on the dynamics of the spins under study and to limit ourselves to considering only its secular part.

Subsequently (see, for example, Graf , 1998; Dollase et al., 2001; Saalwächter , 2002a, 2002b, 2007; Saalwächter et al., 2003, 2006; Fechete, et al., 2002; Mordvinkin and Saalwächter, 2017; Mordvinkin et al., 2020; Vaca Chavez and Saalwächter, 2011; Shahsavan et al., 2022, and references therein), the method was phenomenologically generalized for the

case when the relative spatial displacements of spins during the experiment cannot be considered small, but the situation is such that $T_2^{eff} \ll T_1$. In recent work (Brekotkin et al., 2022) it has been proven by methods of statistical physics that in the limit $\Delta \to 0$ , where $\Delta$  is the time interval between the nearest RF pulses, the phenomenological method of accounting for spatial displacements of spins during the experiment gives correct results and the general relation allowing for quantitative accounting of the corresponding corrections in case of necessity is obtained. .  Interesting analytical results related to DQ

NMR have recently been obtained for model solid-state many-spin one-dimensional I=1/2 systems in which magnetic dipole-dipole interactions have been considered only between nearest neighbors (see Bochkin et al., 2022; Bochkin et al., 2024; Fel'dman et al., 2022, and the literature cited therein).

For the case of the spin system $I = 1/2$, the dominant interactions determining the spin relaxation are, as a rule, magnetic

dipole-dipole interactions. The secular part of the Hamiltonian has the following form (see for example refs. Fatkullin et al., 2012; Fatkullin et al., 2013):

$$\hat{H}_{dd}^{sec} = \sum_{i<j} \hbar\omega_{ij} \left( 2\hat{I}_i^z\hat{I}_j^z - \hat{I}_i^x\hat{I}_j^x - \hat{I}_i^y\hat{I}_j^y \right) = \sum_{i<j} \hbar\omega_{ij} \left( 2\hat{I}_i^z\hat{I}_j^z - \frac{1}{2}\left( \hat{I}_i^+\hat{I}_j^- + \hat{I}_i^-\hat{I}_j^+ \right) \right),$$ (1)

where $\hat{I}_k^+ = \hat{I}_k^x + i\hat{I}_k^y$ , $\hat{I}_k^+ = \hat{I}_k^x + i\hat{I}_k^y$ $\hat{I}_k^- = \hat{I}_k^x - i\hat{I}_k^y$ . The parameter $\omega_{ij}$ describes in frequency units the effective strength of the dipole-dipole coupling of spins with numbers $i$  and j. It is given by the following expression:

$$\omega_{ij} = \frac{1}{8\pi} \frac{\mu_0\gamma^2\hbar\left(1 - 3\cos^2\theta_{ij}\right)}{r_{ij}^3} = D^{ij} \frac{\left(1 - 3\cos^2\theta_{ij}\right)}{2} ,$$ (2)

where $r_{ij}$ is the distance between interacting spins, $\theta_{ij}$ is the angle between direction $Z$ , defined as the direction along which the external magnetic field is aligned, and the vector connecting the discussed spins, $\hat{I}_i^\alpha$ is the operator of the $\alpha$ component of the spin with number $i$ , $\hbar$ is Planck's constant divided by $2\pi$ , $\mu_0$ the magnetic field constant and $\gamma$ is the gyromagnetic ratio of the spins. $D^{ij} = \mu_0\gamma^2\hbar / \left(4\pi r_{ij}^3\right)$  is the dipole-dipole coupling constant. The Hamiltonian (1) in this

paper plays the role of the original spin-lattice interaction Hamiltonian. In the lowest order of perturbation theory, it induces in a spin system 0-quantum transitions (dephasing processes or 0-quantum coherence) by terms  proportional to $\hat{I}_i^z\hat{I}_j^z$ , 1-





quantum transitions (transitions with a flip of one of a pair of interacting spins or 1-quantum coherence) and 2-quantum transitions (coordinated transitions of two spins or 2-quntum coherence) by terms proportional $\hat{I}_i^x \hat{I}_j^x + \hat{I}_i^y \hat{I}_j^y$ .

Under irradiation of the spin system by a special RF pulse sequence referred to as Baum-Pines (BP) sequence (see details in Baum et al., 1985; Baum and Pines, 1986), the dynamics of the spin system at times $t \ll T_1$ is determined not only by the Hamiltonian (1), but also by the effects associated with the irradiation. In fact, there are two conceptually different experiments, each consisting of two stages of equal time duration $\tau_{DQ}$. The first half of both experiments is called the excitation stage, and the second half is called the reconversion stage. At the moment of time $2\tau_{DQ}$ the signal of the studied

spin system is measured. Normally, a 4-step phase cycle is applied to the relative overall phase of the reconversion stage in combination with the receiver phase to filter for either (4n+2)-quantum coherences ("DQ signal") or 4n-quantum coherences ("reference signal"). In the following, we use a simplified yet equivalent description of these two experiments (Saalwächter, 2014). They essentially differ  from each other by the fact that in the first case during both periods of the experiment the phase of RF exposure does not change, and in the second experiment during the period of reconversion the phase of RF

exposure changes by $90^{°}$, which changes a sign of the of the resulting effective spin Hamiltonian, i.e. performs the time-reversal operation with respect to the spin variables.  We will denote the measured signal in the first experiment by $A_0\left(2\tau_{DQ}\right)$, and in the second experiment by $A_1\left(2\tau_{DQ}\right)$.

    In solids, the joint effect of mentioned factors, in the limit $\Delta \to 0$, where $\Delta$ is time interval between the nearest RF pulses, it

is possible to describe the spin system's time evolution in terms of an effective DQ Hamiltonian having the following structure:

$$\hat{H}_{DQ}^n = (-1)^{n\theta\left(t-\tau_{DQ}\right)} \sum_{i<j} \hbar\omega_{ij} \left( \hat{I}_i^y \hat{I}_j^y - \hat{I}_i^x \hat{I}_j^x \right) = (-1)^{n\theta\left(t-\tau_{DQ}\right)} \sum_{i<j} \frac{\hbar\omega_{ij}}{2} \left( \hat{I}_i^+ \hat{I}_j^+ + \hat{I}_i^- \hat{I}_j^- \right), \tag{3}$$

where $\theta\left(x\right)$ is the Heaviside step function, $n = 0,1$. Note, that $n = 0$ corresponds to the first mentioned version of DQ experiment no phase change and $n = 1$ to the second version 90° phase shift.


    The Hamiltonian (3), in contrast to (1), induces in the lowest order of perturbation theory, or times $t \ll T_{fl}^*$, where $T_{fl}^*$ is the characteristic time of flip-flop processes created by Hamiltonian (1), only DQ transitions (or creates DQ coherences) of interacting spins. At longer times, of course more complex quantum transitions involving coherent behavior of even spins become essential. At time moment $t = \tau_{DQ}$ the operator (3) changes time for the case, when $n = 1$.





For situations where spins are moving, expression (3) was heuristically generalized by considering the coupling constant as a function of time. The validity of such a generalization is shown in ref. (Brekotkin et. al., 2022) by a sequential quantum-statistical calculation showing that the terms added to the relation vanish in the limit $\Delta \to 0$; in addition, a general relation is obtained that allows one to quantify the corresponding contributions to the experimentally measured signal, if necessary. In this paper we will ignore the mentioned effect and work with an effective Hamiltonian of mutual action of the form:


$$\hat{H}_{DQ}^{n}\left(t\right) = \left(-1\right)^{n\theta\left(t-\tau_{DQ}\right)} \sum_{i<j} \hbar\omega_{ij}^{(n)}\left(t\right)\left(\hat{I}_{i}^{y}\hat{I}_{j}^{y} - \hat{I}_{i}^{x}\hat{I}_{j}^{x}\right),$$ (4)

where

$$\omega_{ij}^{(n)}\left(t\right) = \left(-1\right)^{n\theta\left(t-\tau_{DQ}\right)} \omega_{ij}\left(t\right) \qquad .$$ (5)

Here, $\omega_{ij}\left(t\right)$ can be obtained from $\omega_{ij}$ of (1) and (3) by transition to the ordinary quantum mechanical interaction (Dirac)

representation, where the role of the zero Hamiltonian is sum of the lattice Hamiltonian and the Hamiltonian of Zeeman interaction of investigated spins with an external magnetic field. Recall that the properties of the Heaviside step function used in eq. (5) are such that it $\theta\left(x\right)=0$ at $x<0$ and $\theta\left(x\right)=1$ at $x \geq 0$, which allows us to account analytically for the inverted sign of the DQ Hamiltonian during the reconversion period in experiments when n=1.

The Hamiltonian (4) itself is already quite complex and does not allow for accurate calculations in nontrivial cases. The standard approximation that allows one to obtain closed analytic relations is the Anderson-Weiss approximation, the second cumulant approximation in common parlance, which completely ignores the effects of flip-flop processes. Meanwhile, the Hamiltonian (4) can be rewritten in the form:

$$\hat{H}_{DQ}^{(n)}\left(t\right) = \sum_{i<j} 2\hbar\omega_{ij}^{(n)}\left(t\right)\hat{I}_{i}^{y}\hat{I}_{j}^{y} - \sum_{i<j} \hbar\omega_{ij}^{(n)}\left(t\right)\left(\hat{I}_{i}^{y}\hat{I}_{j}^{y} + \hat{I}_{i}^{x}\hat{I}_{j}^{x}\right)$$
$$= \sum_{i<j} 2\hbar\omega_{ij}^{(n)}\left(t\right)\hat{I}_{i}^{y}\hat{I}_{j}^{y} - \frac{1}{2}\sum_{i<j} \hbar\omega_{ij}^{(n)}\left(t\right)\left(\hat{I}_{i}^{+}\hat{I}_{j}^{-} + \hat{I}_{i}^{-}\hat{I}_{j}^{+}\right)$$ (6)

By direct calculation one can see that:

$$\left[\hat{I}^{z}; \sum_{i<j} \hbar\omega_{ij}^{(n)}\left(t\right)\left(\hat{I}_{i}^{y}\hat{I}_{j}^{y} + \hat{I}_{i}^{x}\hat{I}_{j}^{x}\right)\right] = \frac{1}{2}\sum_{i<j} \hbar\omega_{ij}^{(n)}\left(t\right)\left[\hat{I}_{z}; \hat{I}_{i}^{+}\hat{I}_{j}^{-} + \hat{I}_{i}^{-}\hat{I}_{j}^{+}\right] = 0,$$ (7)

where $\hat{I}^{z} = \sum_{k} \hat{I}_{k}^{z}$ is z-component of the total spin system.



This exact result allows us to modify the usual Anderson-Weiss approximation so that the effects of flip-flop processes will

be accounted for on the experimentally observed signals at least in the mean-field approximation. A detailed study of this

circumstance is the main purpose of this article.


## 2 Theoretical part.

### 2.1 General consideration.

An inevitable and important initial element of all quantum-statistical calculations of experimentally measured dynamical

quantities is the transition to the interaction representation or, which is synonymous, to the Dirac representation. This

transition is performed by dividing the full initial Hamiltonian of the system $\hat{H}$ into a sum of the "zero" Hamiltonian $\hat{H}_0$

and the "interaction" Hamiltonian $\hat{H}_{int}$ :

$$\hat{H} = \hat{H}_0 + \hat{H}_{int} \ . \tag{8}$$

In problems of NMR spectroscopy, the lattice Hamiltonian $\hat{H}_L$ and the Hamiltonian of the Zeeman interaction of the spins

under study ($\hat{H}_Z = \hbar\omega_0 \hat{I}_z$, $\omega_0$ being the resonance frequency), are usually included in $\hat{H}_0 = \hat{H}_L + \hat{H}_Z$ ; all other interactions

are assumed to be included in $\hat{H}_{int}$ . In the DQ experiments discussed in this paper, the measured quantity is the z-

component of the total system spin $\hat{I}^z = \sum_k \hat{I}_k^z$ . The existence of the identity (7) allows one to include the DQ part of the

Hamiltonian

$$\hat{H}_{DQ}^{fl} \equiv -\sum_{i<j} \hbar\omega_{ij}^{(n)}(t) \left( \hat{I}_i^y \hat{I}_j^y + \hat{I}_i^x \hat{I}_j^x \right) \tag{9}$$

in the zero part of the Hamiltonian, thereby reformulating the transition into the Dirac representation. Only such a procedure

should be accurately described, since the initial Hamiltonian itself (3) is an effective Hamiltonian generated by the joint

action on the spin system of the secular part of the Hamiltonian of magnetic dipole-dipole interactions (1) and the irradiation

of the RF spin system by the Pines-Baum sequence.

In the laboratory coordinate system and Schrödinger representation in the DQ resonance experiments due to the irradiation

of the spin system of the BP sequence, or its equivalent modifications, we have an initially time-dependent Hamiltonian:





$$\hat{H}^{(n)}(t) = \hat{H}_L + \hat{H}_Z + \hat{H}_{dd}^{sec} + \hat{H}_{BP}^{(n)}(t), \tag{10}$$

where $\hat{H}_{BP}^{(n)}(t)$ is the Hamiltonian of the interaction of the studied spin system with the irradiation field of BP pulse sequence of type $n = 0,1$. Consequently, the evolution operator is initially a Dyson chronological, time- ordered, exponent:

$$\hat{U}^{(DQ,n)}(t) = \hat{T} \exp \left\{ -\frac{i}{\hbar} \int_0^t \hat{H}^{(n)}(t_1) dt_1 \right\}. \tag{11}$$

The transition to the interaction representation will be carried out in two steps. At the first stage the role of the zero Hamiltonian is played as usual the following Hamiltonian:

$$\hat{H}_0(t) = \hat{H}_L + \hat{H}_Z. \tag{12}$$

After that the relation (11) can be rewritten as follows:

$$\hat{U}^{(DQ,n)}(t) = \hat{U}_0(t)\hat{U}_1^{(DQ,n)}(t), \tag{13}$$

where

$$\hat{U}_0(t) = \hat{T} \exp\left\{ -\frac{i}{\hbar}\int_0^t \hat{H}_0 dt_1 \right\} = \exp\left\{ -\frac{i}{\hbar}\hat{H}_0 t \right\},$$

$$\hat{U}_1^{(DQ,n)}(t) = \hat{T} \exp\left\{ -\frac{i}{\hbar}\int_0^t \left( \hat{H}_{dd}^{sec(n)}(t_2) + \hat{\tilde{H}}_{BP}^{(n)}(t_2) \right) dt_2 \right\}, \tag{14}$$

$$\hat{H}_{dd}^{sec(n)}(t_2) + \hat{\tilde{H}}_{BP}^{(n)}(t_2) = \left( \hat{U}_0^{(DQ,n)}(t_2) \right)^{-1} \left( \hat{H}_{dd}^{sec} + \hat{H}_{BP}^{(n)}(t_2) \right) \hat{U}_0^{(DQ,n)}(t_2)$$

In the limit $\Delta \to 0$, according to ref. (Brekotkin et. al., 2022) , we have

$$\hat{H}_{dd}^{sec(n)}(t_2) + \hat{\tilde{H}}_{BP}^{(n)}(t_2) \xrightarrow{\Delta \to 0} \hat{H}_{DQ}^{(0)}(t_2) = \sum_{i<j} \hbar \omega_{ij}^{(n)}(t_2) \left( \hat{I}_i^y \hat{I}_j^y - \hat{I}_i^x \hat{I}_j^x \right). \tag{15}$$

Now we can return to the relation (6) and introduce the following notations to shorten the formulas:

$$\hat{H}_{DQ}^{(n)}(t) = \hat{H}_{DQ,yy}^{(n)}(t) + \hat{H}_{DQ,fl}^{(n)}(t), \tag{16}$$

where

$$\hat{H}_{DQ,yy}^{(n)}(t) = \sum_{i<j} 2\hbar \omega_{ij}^{(n)}(t) \hat{I}_i^y \hat{I}_j^y \tag{17}$$

and

$$\hat{H}_{DQ,fl}^{(n)}(t) = -\sum_{i<j} \hbar \omega_{ij}^{(n)}(t) \left( \hat{I}_i^y \hat{I}_j^y + \hat{I}_i^x \hat{I}_j^x \right). \tag{18}$$






In DQ experiments, the measured quantity is the z-component of the total magnetic moment of the resonant spins, which in turn is proportional to the z-component of the total spin $\hat{I}^z = \sum_k \hat{I}_k^z$. The measurement, as already noted, is carried out for a time $t = 2\tau_{DQ}$. In the high-temperature approximation by spin variables, the signal with $n = 0,1$ for a given spin system is

$$A_n(2\tau_{DQ}) = \frac{\beta\hbar\omega_0}{(2I+1)^{N_s}}\left\langle Tr_s\left(\hat{I}^z\hat{U}^{(DQ,n)}(2\tau_{DQ})\hat{I}^z\hat{U}^{*(DQ,n)}(2\tau_{DQ})\right)\right\rangle_{eq},\tag{19}$$

where $N_s$ is the total number of spins in the system with the resonance frequency $\omega_0$, $\beta$ is the inverse temperature, and the trace operation $Tr_s(...)$ is performed over the spin variables, and due to the unitarity of the propagator one has $\hat{U}^{*(DQ,n)}(t) = \left(\hat{U}^{(DQ,n)}(t)\right)^{-1}$ and bracket $\langle...\rangle_{eq}$ denotes equilibrium averaging over all lattice variables.

Note once again that $\left[\hat{I}_z; \hat{H}_0\right] = 0$, which allows to rewrite the expression (19) as follows:

$$A_n(2\tau_{DQ}) = \frac{\beta\hbar\omega_0}{(2I+1)^{N_s}}\left\langle Tr_s\left(\hat{I}^z\hat{U}_1^{(DQ,n)}(2\tau_{DQ})\hat{I}^z\hat{U}_1^{*(DQ,n)}(2\tau_{DQ})\right)\right\rangle_{eq}.\tag{20}$$

Next, we represent the propagator $\hat{U}_1^{(DQ,n)}(t)$ as:

$$\hat{U}_1^{(DQ,n)}(t) = \left(\hat{T}\exp\left(-\frac{i}{\hbar}\int_0^t \hat{H}_{DQ,fl}^{(n)}(t_1)dt_1\right)\right)\left(\hat{T}\exp\left(-\frac{i}{\hbar}\int_0^t \hat{\bar{H}}_{DQ,yy}^{(n)}(t_1)dt_1\right)\right),\tag{21}$$

where

$$\hat{\bar{H}}_{DQ,yy}^{(n)}(t) = \left(\hat{T}\exp\left(-\frac{i}{\hbar}\int_0^t \hat{H}_{DQ,fl}^{(n)}(t_1)dt_1\right)\right)^{-1}\hat{H}_{DQ,yy}^{(n)}(t)\left(\hat{T}\exp\left(-\frac{i}{\hbar}\int_0^t \hat{H}_{DQ,fl}^{(n)}(t_1)dt_1\right)\right).\tag{22}$$

It is worth noting that the transition reflected in expressions (21) and (22) is analogous to the transition in (13) and (14), and represents the second step in transitioning to the representation of interaction that interests us. Substituting the relation (22) into (20), taking into account the identity (7), we obtain:

$$A_n(2\tau_{DQ}) = \frac{\beta\hbar\omega_0}{(2I+1)^{N_s}}\left\langle Tr_s\left(\hat{I}^z\hat{U}_2^{(DQ,n)}(2\tau_{DQ})\hat{I}^z\hat{U}_2^{*(DQ,n)}(2\tau_{DQ})\right)\right\rangle_{eq},\tag{23}$$

where

$$\hat{U}_2^{(DQ,n)}(t) = \hat{T}\exp\left(-\frac{i}{\hbar}\int_0^t \hat{\bar{H}}_{DQ,yy}^{(n)}(t_1)dt_1\right)\tag{24}$$

Note that in deriving relations (22) and (23), only a single asymptotically exact approximation (15) is made.



## 2.2 The simplest approximation.

If we neglect the influence of flip-flop processes, i.e., put $\hat{H}_{DQ,fl}^{(n)}(t) = 0$ in the relations (21) and (22), then with respect to

the spin variables, the relation (23) can be counted exactly, if dynamics of the lattice variables can be treated classically.

Indeed, in this case $\left[ \hat{H}_{DQ,yy}^{(n)}(t_2); \hat{H}_{DQ,yy}^{(n)}(t_1) \right] = 0$ and the propagator $\hat{U}_2^{(DQ,n)}(t)$ with respect to spin variables take a

relatively simple form:

$$\hat{U}_2^{(DQ,n)}(t) \simeq \exp\left( -\frac{i}{\hbar} \int_0^t \hat{\hat{H}}_{DQ,yy}^{(n)}(t_1) dt_1 \right) = \exp\left( -i \sum_{i<j} 2\varphi_{ij}^{(n)}(t) \hat{I}_i^y \hat{I}_j^y \right) , \qquad (25)$$

where

$$\varphi_{ij}^{(n)}(t) = \int_0^t \omega_{ij}^{(n)}(t_1) dt_1 \quad . \qquad (26)$$

Substituting relations (25) and (26) into formula (20), using the algebraic properties of spin operators for I=1/2, see details in (Fatkullin et al., 2012; Fatkullin et al., 2013), we obtain the following simplest approximation for experimentally observed DQ signals:

$$A_n^{sim}(2\tau_{DQ}) = \frac{\beta\hbar\omega_0}{4} \sum_k \left\langle \prod_i \cos\left( \varphi_{ik}^{ex} + (-1)^n \varphi_{ik}^{rec} \right) \right\rangle_{eq}$$

$$I_{nDQ}^{sim}(\tau_{DQ}) \equiv \frac{1}{2} \left( \frac{A_1(2\tau_{DQ}) - A_0(2\tau_{DQ})}{A_1(2\tau_{DQ})} \right) \qquad , \qquad (27)$$

$$= \frac{1}{2} \left( 1 - \frac{\sum_i \left\langle \prod_j \cos\left( \varphi_{ij}^{ex} + \varphi_{ij}^{rec} \right) \right\rangle_{eq}}{\sum_i \left\langle \prod_j \cos\left( \varphi_{ij}^{ex} - \varphi_{ij}^{rec} \right) \right\rangle_{eq}} \right)$$

with

$$\varphi_{ik}^{ex} = \int_0^{\tau_{DQ}} \omega_{ik}(t_1) dt_1 \quad , \quad \varphi_{ik}^{rec} = \int_{\tau_{DQ}}^{2\tau_{DQ}} \omega_{ik}(t_1) dt_1 \quad . \qquad (28)$$

It is important to note that the derivation of relation (27) proposed in this paper, in contrast to the previous work (Fatkullin et al., 2013) that does not use the Anderson-Weiss approximation directly. For a system of spin pairs, when $i, k = 1, 2$, our expression (27) exactly recovers the known result:

$$A_n^{pair}(2\tau_{DQ}) = \frac{\beta\hbar\omega_0}{2} \left\langle \cos\left( \varphi_{12}^{ex} + (-1)^n \varphi_{12}^{rec} \right) \right\rangle_{eq} . \qquad (29)$$

## 2.3 Flip-flop transitions effect.



Flip-flop transitions, as already noted, are induced by the partial Hamiltonian $\hat{H}_{DQ,fl}^{(n)}(t)$ determined by the expression (18).

Their influence on the experimentally observed signals $A_n(2\tau_{DQ})$ is through the Hamiltonian $\hat{\tilde{H}}_{DQ,yy}^{(n)}(t)$, see relations
(20)-(22), which can be rewritten in the following way:

$$\hat{\tilde{H}}_{DQ,yy}^{(n)}(t) = \sum_{i<j} 2\hbar\omega_{ij}^{(n)}(t)\left(\hat{I}_i^y\hat{I}_j^y\right)_t^{fl}, \tag{30}$$

where

$$\left(\hat{I}_i^y\hat{I}_j^y\right)_t^{fl} \equiv \left(\hat{T}\exp\left(-\frac{i}{\hbar}\int_0^t \hat{H}_{DQ,fl}^{(n)}(t_1)\,dt_1\right)\right)^{-1} \hat{I}_i^y\hat{I}_j^y \left(\hat{T}\exp\left(-\frac{i}{\hbar}\int_0^t \hat{H}_{DQ,fl}^{(n)}(t_1)\,dt_1\right)\right). \tag{31}$$

The approximation considered earlier in Section 2.b is equivalent to neglecting the dependence of operators $\left(\hat{I}_i^y\hat{I}_j^y\right)_t^{fl}$ on

time. In this section we will consider the approximation by its projection in Liouville spin space to the initial value:

$$\left\{\hat{I}_i^y\hat{I}_j^y\right\}_t^{fl} \simeq \hat{I}_i^y\hat{I}_j^y \frac{Tr_s\left(\hat{I}_i^y\hat{I}_j^y\left\{\hat{\tilde{I}}_i^y\hat{\tilde{I}}_j^y\right\}_t^{fl}\right)}{Tr_s\left(\left(\hat{I}_i^y\right)^2\left(\hat{I}_j^y\right)^2\right)} \equiv \tilde{P}_{ij}^{n,fl}\{t;0\}\,\hat{I}_i^y\hat{I}_j^y. \tag{32}$$

The quantity $\tilde{P}_{ij}^{n,fl}\{t;0\}$ does not depend on spin variables, but it is a complex function of time dependent lattice variables.
Later we will see that after averaging over the lattice variables, it can be for the case of $n=0$ viewed as the probability that
during the time interval $t$ none of the spins in question with numbers i and j participated in the flip-flop process with another
spins.

Consider the expansion of the value $\tilde{P}_{ij}^{n,fl}\{t_2;t_1\}$ in a perturbation theory series with respect to $\hat{H}_{DQ,fl}^{(n)}(t_1)$:

$$\tilde{P}_{ij}^{n,fl}\{t_2;t_1\} = 1 - \frac{i}{2}\int_{t_1}^{t_2} d\tau_1 \sum_{k,l} \omega_{kl}^{(n)}(\tau_1) \frac{Tr_s\left(\hat{I}_i^y\hat{I}_j^y\left[\hat{I}_k^x\hat{I}_l^y + \hat{I}_k^y\hat{I}_l^x;\hat{I}_i^y\hat{I}_j^y\right]\right)}{Tr_s\left(\left(\hat{I}_i^y\right)^2\left(\hat{I}_j^y\right)^2\right)} -$$
$$- \frac{1}{4}\int_{t_1}^{t_2} d\tau_2 \int_{t_1}^{\tau_2} d\tau_1 \sum_{k,l;s,t} \omega_{kl}^{(n)}(\tau_2)\omega_{st}^{(n)}(\tau_1) \frac{Tr_s\left(\hat{I}_i^y\hat{I}_j^y\left[\hat{I}_k^x\hat{I}_l^x + \hat{I}_k^y\hat{I}_l^y;\left[\hat{I}_s^x\hat{I}_t^x + \hat{I}_s^y\hat{I}_t^y;\hat{I}_i^y\hat{I}_j^y\right]\right]\right)}{Tr_s\left(\left(\hat{I}_i^y\right)^2\left(\hat{I}_j^y\right)^2\right)} + ... \tag{33}$$



The first-order contribution by $\omega_{kl}^{(n)}(\tau_1)$ in the relation (33) turns out to be exactly 0:

$$
\begin{aligned}
Tr_s\left(\hat{I}_i^y\hat{I}_j^y\left[\hat{I}_k^x\hat{I}_l^x+\hat{I}_k^y\hat{I}_l^y;\hat{I}_i^y\hat{I}_j^y\right]\right) &= -Tr_s\left(\hat{I}_i^y\hat{I}_j^y\left[\hat{I}_i^y\hat{I}_j^y;\hat{I}_k^x\hat{I}_l^x+\hat{I}_k^y\hat{I}_l^y\right]\right)= \\
&= Tr_s\left(\left[\hat{I}_i^y\hat{I}_j^y;\hat{I}_i^y\hat{I}_j^y\right];\hat{I}_k^x\hat{I}_l^x+\hat{I}_k^y\hat{I}_l^y\right)=0
\end{aligned}
\tag{34}
$$

Then relation is thus simplified:

$$
\tilde{P}_{ij}^{(n),fl}(t_2;t_1)=1-\frac{1}{4}\int_{t_1}^{t_2}d\tau_2\int_{t_1}^{\tau_2}d\tau_1\sum_{k,l;s,t}\omega_{kl}^{(n)}(\tau_2)\omega_{st}^{(n)}(\tau_1)\frac{Tr_s\left(\left[\hat{I}_i^y\hat{I}_j^y;\hat{I}_k^x\hat{I}_l^x\right]\left[\hat{I}_s^x\hat{I}_t^x;\hat{I}_i^y\hat{I}_j^y\right]\right)}{Tr_s\left(\left(\hat{I}_i^y\right)^2\left(\hat{I}_j^y\right)^2\right)}+\dots
\tag{35}
$$

The standard commutator and trace calculations on the right-hand side lead to the following result:

$$
\begin{aligned}
&\sum_{k,l;s,t}\omega_{kl}^{(n)}(\tau_2)\omega_{st}^{(n)}(\tau_1)\frac{Tr_s\left(\left[\hat{I}_i^y\hat{I}_j^y;\hat{I}_k^x\hat{I}_l^x\right]\left[\hat{I}_s^x\hat{I}_t^x;\hat{I}_i^y\hat{I}_j^y\right]\right)}{Tr_s\left(\left(\hat{I}_i^y\right)^2\left(\hat{I}_j^y\right)^2\right)} \\
&= \frac{4}{3}I(I+1)\sum_k{}'\left(\omega_{ik}^{(n)}(\tau_2)\omega_{ik}^{(n)}(\tau_1)+\omega_{jk}^{(n)}(\tau_2)\omega_{jk}^{(n)}(\tau_1)\right)+\frac{8}{5}\left\{I(I+1)-\frac{3}{4}\right\}\omega_{ij}^{(n)}(\tau_2)\omega_{ij}^{(n)}(\tau_1)
\end{aligned}
\tag{36}
$$

where $\sum_k{}'\dots$ means, that summation is performed with restrictions $k\neq i,j$.

For the case of spins $I=1/2$, the last term in the right-hand side of relation (36) is exactly 0. This is expected because the mutual flip-flop transitions between spins with numbers i and j do not change the value of the product $\hat{I}_i^y\hat{I}_j^y$ in the case under consideration. At the same time, flip-flop transitions with other system spins accounted for in the terms of the proportional to $\omega_{ik}^{(n)}(\tau_2)\omega_{ik}^{(n)}(\tau_1)+\omega_{jk}^{(n)}(\tau_2)\omega_{jk}^{(n)}(\tau_1)$ change the value of $\hat{I}_i^y\hat{I}_j^y$.

Hereafter, we will assume that the motion of the lattice variables is correctly described by classical dynamics, which allows us to neglect the time ordering of the corresponding variables; the relation (35) takes the following form:

$$
\begin{aligned}
\tilde{P}_{ij}^{n,fl}\{t_2;t_1\} &= 1-\frac{I(I+1)}{3}\int_{t_1}^{t_2}d\tau_2\int_{t_1}^{\tau_2}d\tau_1\sum_k{}'\left(\omega_{ik}^{(n)}(\tau_2)\omega_{ik}^{(n)}(\tau_1)+\omega_{jk}^{(n)}(\tau_2)\omega_{jk}^{(n)}(\tau_1)\right)+\dots \\
&= 1-\frac{1}{6}I(I+1)\sum_k{}'\left(\left(\varphi_{ik}^{(n)}(t_2;t_1)\right)^2+\left(\varphi_{jk}^{(n)}(t_2;t_1)\right)^2\right)+\dots
\end{aligned}
\tag{37}
$$

where $\varphi_{st}^{(n)}(t_2;t_1)=\int_{t_1}^{t_2}\omega_{st}^{(n)}(\tau)d\tau$.





Note that in the right-hand side of (32) we have taken into account only projections of the operator $\left(\hat{I}_i^y \hat{I}_j^y\right)_t^{fl}$ on the initial

spin operator $\hat{I}_i^y \hat{I}_j^y$ and neglected by its projections on the spin operators like $\hat{I}_k^y \hat{I}_l^y$ for the cases when $k \neq i, j$ or $l \neq i, j$. By

direct and lengthy calculations analogous to (33)-(36) one can see that in the second order of perturbation theory by $\omega_{kl}^{(n)}(\tau)$

, the contributions from them into the expression (37) are exactly zero.


In further calculations, we will apply the Anderson-Weiss approximation with respect to the magnitude $\tilde{P}_{ij}^{n,fl}\{t_2; t_1\}$ :

$$\tilde{P}_{ij}^{n,fl}\{t_2; t_1\} = \exp\left\{-\frac{1}{6}I(I+1)\sum_k{}' \left(\left(\varphi_{ik}^{(n)}(t_2; t_1)\right)^2 + \left(\varphi_{jk}^{(n)}(t_2; t_1)\right)^2\right)\right\}. \tag{38}$$

It seems appropriate to note that the approximation (32) reconstructs the formal mathematical structure of the previously

studied Hamiltonian (16) by modifying in it only the time-dependent spin-lattice interaction constant:

$$\hat{\tilde{H}}_{DQ,yy}^{(n)}(t) \simeq \sum_{i<j} 2\hbar\tilde{\omega}_{ij}^{(n)}(t)\hat{I}_i^y \hat{I}_j^y \;, \tag{39}$$

with

$$\tilde{\omega}_{ij}^{(n)}(t) = \tilde{P}_{ij}^{n,fl}(t;0)\omega_{ij}^{(n)}(t). \tag{40}$$

Effects associated with the Hamiltonian (18) are now considered by the presence of a multiplier $\tilde{P}_{ij}^{fl}(t;0)$. Therefore, by

analogy with relation (27), we can immediately write down the relations for experimentally observed signals:

$$A_n(2\tau_{DQ}) = \frac{\beta\hbar\omega_0}{4}\sum_k \left\langle\prod_i \cos\left(\tilde{\varphi}_{ik}^{ex} + (-1)^n \tilde{\varphi}_{ik}^{rec}\right)\right\rangle_{eq}, \tag{41}$$

with

$$\tilde{\varphi}_{ik}^{ex} = \int_0^{\tau_{DQ}} \tilde{\omega}_{ik}(t_1)\,dt_1 \quad, \quad \tilde{\varphi}_{ik}^{rec} = \int_{\tau_{DQ}}^{2\tau_{DQ}} \tilde{\omega}_{ik}(t_1)\,dt_1 \quad. \tag{42}$$

From the experimentally measured quantities $A_0(2\tau_{DQ})$ and $A_1(2\tau_{DQ})$ one can construct the so-called normalized DQ

build up function:

$$I_{nDQ}(\tau_{DQ}) = \frac{1}{2}\frac{A_1(2\tau_{DQ}) - A_0(2\tau_{DQ})}{A_1(2\tau_{DQ})}. \tag{43}$$

Using the expression (41) we get:





$$I_{nDQ}\left(\tau_{DQ}\right) = \frac{1}{2}\left(1 - \frac{\sum_i \left\langle \prod_j \cos\left(\tilde{\varphi}_{ij}^{(0),ex} + \tilde{\varphi}_{ij}^{(0),rec}\right)\right\rangle_{eq}}{\sum_i \left\langle \prod_j \cos\left(\tilde{\varphi}_{ij}^{(1),ex} - \tilde{\varphi}_{ij}^{(1),rec}\right)\right\rangle_{eq}}\right).$$

(44)


### 3    Discussion.

Relations (41)-(44) are the main general results of this paper. They have a formal mathematical structure with a more
simplified approach, see relation (27), completely neglecting flip-flops during DC experiments. The difference is hidden in

the values $\varphi_{ij}^{(n)} = \int_0^{2\tau_{DQ}} \omega_{ij}^{(n)}(t_1)dt_1$ in expression (27)   and $\tilde{\varphi}_{ij}^{(n)} = \int_0^{2\tau_{DQ}} \tilde{\omega}_{ij}^{(n)}(t_1)dt_1$, with   $\tilde{\omega}_{ij}^{(n)}(t) = \tilde{P}_{ij}^{n,fl}(t;0)\omega_{ij}^{(n)}(t)$, in the

expressions (41)-(44). As we can see, the influence of flip-flop processes was taken into account by multiplying the

frequencies $\omega_{ij}^{(n)}(t_1)$ by the values $\tilde{P}_{ij}^{n,fl}(t;0)$, defined by the approximation (32). From this definition, it is natural to

expect that the quantities discussed $\tilde{P}_{ij}^{n,fl}(t)$ should be closely related to the probabilities of spins with numbers i and j
during the time interval t not to participate in flip-flop processes with any other spin of the system. Let's start with a more
detailed discussion of this connection.

### 3.1 Interpretation of  $\tilde{P}_{ij}^{n,fl}\{t_2;t_1\}$.

The quantity under consideration is defined by relation (32), expression (37) is its initial Taylor series expansion in
$\hat{H}_{DQ,fl}^{(n)}(t)$, and (38) is similarity of its Anderson-Weiss approximation. Averaging in all the indicated relations was carried

out only over spin variables. Therefore, according to formula (31) $\tilde{P}_{ij}^{n,fl}\{t_2;t_1\}$  remains a function of lattice variables, i.e., it

depends, in general, on the spatial coordinates of spins at all previous time moments. It is not in itself experimentally

measurable. Experimentally measured quantities that depend on $\tilde{P}_{ij}^{n,fl}\{t_2;t_1\}$ , as follows from formulas (19), (43), (44),
contain an additional averaging over the equilibrium distribution of lattice variables.   In mathematical terms, the quantity

$\tilde{P}_{ij}^{n,fl}\{t_2;t_1\}$ is a multidimensional random process, microscopically defined by the Hamiltonian of lattice variables $\hat{H}_L$.





If we average the expression (37) over the lattice variables, we obtain:

$$
P_{ij}^{n,fl}\{t_2;t_1\} \equiv \left\langle \tilde{P}_{ij}^{n,fl}\{t_2;t_1\}\right\rangle_{eq}
$$

$$
=1-\frac{I(I+1)}{3}\int_{t_1}^{t_2}d\tau_2\int_{t_1}^{\tau_2}d\tau_1\sum_k{}'\left(\left\langle \omega_{ik}^{(n)}(\tau_2)\omega_{ik}^{(n)}(\tau_1)+\omega_{jk}^{(n)}(\tau_2)\omega_{jk}^{(n)}(\tau_1)\right\rangle_{eq}\right)+\dots. \tag{45}
$$

$$
=1-\frac{1}{6}I(I+1)\sum_k{}'\left(\left\langle\left(\varphi_{ik}^{(n)}(t_2;t_1)\right)^2\right\rangle_{eq}+\left\langle\left(\varphi_{jk}^{(n)}(t_2;t_1)\right)^2\right\rangle_{eq}\right)+\dots
$$

Using the Anderson-Weiss approximation for this relation, we obtain:

$$
P_{ij}^{n,fl}\{t_2;t_1\}=\exp\left\{-\frac{1}{6}I(I+1)\sum_k{}'\left(\left\langle\left(\varphi_{ik}^{(n)}(t_2;t_1)\right)^2\right\rangle_{eq}+\left\langle\left(\varphi_{jk}^{(n)}(t_2;t_1)\right)^2\right\rangle_{eq}\right)\right\}. \tag{46}
$$

An expression similar to eqs. (45,46) with numerical multiplier accuracy was obtained in ref. (Fatkullin et al., 2012), which discussed the modified Anderson-Weiss approximation with respect to the Free Induction Decay signal of the proton spin system:

$$
P_{kl}^{FID,fl}(t)=\exp\left\{-\int_0^t d\tau(t-\tau)\frac{I(I+1)}{6\hbar^2}\sum_m{}'\left(\left\langle\tilde{A}_{km}(t)\tilde{A}_{km}(0)\right\rangle_{eq}+\left\langle\tilde{A}_{lm}(t)\tilde{A}_{lm}(0)\right\rangle_{eq}\right)\right\}\ , \tag{47}
$$

where $\tilde{A}_{kl}(t)=A_{kl}(t)-2J_{kl}(t)$, $J_{kl}$ is the constant of exchange interaction, J-coupling, between spins with numbers $k$ and $l$ ,(misprint with numerical coefficient before exchange constant in (Fatkullin et al., 2012) is corrected) and $A_{kl}(t)=\frac{\gamma^2\hbar^2}{r_{kl}^3(t)}\left(1-3\cos^2\left(\theta_{kl}(t)\right)\right)$ with variables identical to the expression (2) of this paper. The value $P_{kl}^{FID,fl}(t)$ can be viewed as the probability that a given pair of numbered k and l spins will not participate in flip-flop processes over time t 295 with any third spin of the system.

In terms of present paper, $J_{kl}=0$, $A_{kl}(t)=2\omega_{kl}(t)$, if to consider motions of the lattice variables as classical, the expression (47) can be rewritten in the following way:

$$
P_{kl}^{FID,fl}(t)=\exp\left\{-\int_0^t d\tau(t-\tau)\frac{2I(I+1)}{3\hbar^2}\sum_m{}'\left(\left\langle\omega_{km}(\tau)\omega_{km}(0)\right\rangle_{eq}+\left\langle\omega_{lm}(\tau)\omega_{lm}(0)\right\rangle_{eq}\right)\right\}.
$$

$$
=\exp\left\{-\frac{1}{3}I(I+1)\sum_m{}'\left(\left\langle\left(\varphi_{km}^{(0)}(t;0)\right)^2\right\rangle_{eq}+\left\langle\left(\varphi_{lm}^{(0)}(t;0)\right)^2\right\rangle_{eq}\right)\right\} \tag{48}
$$

The difference in the ratios (46) and (48) in the numerical coefficients 1/6 and 1/3, respectively, is noticeable. It is related to 300 the fact that in the first case the flip-flop processes are induced by the modified RF irradiated Hamiltonian (4) and in the second case by the secular part of the Hamiltonian of magnetic dipole-dipole interactions (1). The indicated difference in the numerical coefficients indicates that in the case of DQ experiments, the influence of flip-flop processes is weaker than for





FID or Hahn echo and will appear at later times. To complete the picture, it seems appropriate to quote an expression from ref. (Fatkullin et al., 2012) for the probability that spin number k will not participate in flip-flop processes at intervals t:

$$
\begin{aligned}
P_k^{FID,fl}(t) &= \exp\left\{-\int_0^t d\tau (t-\tau) \frac{I(I+1)}{6\hbar^2} \sum_m {}'\left\langle \tilde{A}_{km}(\tau)\tilde{A}_{km}(0)\right\rangle_{eq}\right\}, \\
&= \exp\left\{-\frac{I(I+1)}{3} \sum_m {}'\left\langle \left(\varphi_{km}^{ex}(t;0)\right)^2\right\rangle_{eq}\right\}
\end{aligned}
\tag{49}
$$

where the second line on the right-hand side is rewritten in terms of the variables of this paper.

From these examples, it seems to us that the value $\tilde{P}_{ij}^{n,fl}\{t_2;t_1\}$ for the case, when $n=0$ can be regarded as a conditional probability for spins with numbers i and j not to participate in flip-flop processes with other system spins during the time interval $t_1 \le t \le t_2$, provided that the lattice variables changed along the phase trajectory defined by a particular set of lattice coordinates during the specified time interval.

### 3.2 The case of a (quasi-)rigid lattice and Anderson-Weiss approximation.

In this case spins make only small oscillations in the vicinity of the equilibrium positions. Alternatively, the case also applies to anisotropic fast-limit motions (such as in polymer networks at high temperatures), where the dipolar couplings are replaced by quasi-static, possibly rather small residual dipolar couplings of order $3\omega_{ij}/(5N)$, where N is the number of statistical segments between crosslinks. Also in this case, relevant oscillations around the mean value of $3\omega_{ij}/(5N)$ are rather small for relevant long timescales. In both cases, the time dependence of frequencies $\omega_{ij}(t)$ in the Hamiltonian (4) can be neglected and consequently $\omega_{ij}^{(n)}(t) = (-1)^{n\theta(t-\tau_{DQ})}\omega_{ij}$. Then, the signal $A_1(t)$ is always fully recovered at time $t = 2\tau_{DQ}$. Our approximation (32) preserves this property. This follows from the relations (37) and (38), since as is easy to see, we have for $0 \le t \le \tau_{DQ}$:

$$
\tilde{P}_{ij}^{1,fl}\{t+\tau_{DQ};0\} = \tilde{P}_{ij}^{1,fl}\{\tau_{DQ}-t;0\}
\tag{50}
$$

and therefore

$$
\tilde{\varphi}_{ij}^{(1),ex} - \tilde{\varphi}_{ij}^{(1),rec} = \omega_{ij}\int_0^{\tau_{DQ}} \left(\tilde{P}_{ij}^{1,fl}\{t_1+\tau_{DQ};0\} - \tilde{P}_{ij}^{1,fl}\{\tau_{DQ}-t_1;0\}\right)dt_1 = 0 .
\tag{51}
$$

Note that this property is not trivial at all, since the flip-flop processes are elementary steps of spin diffusion, which is irreversible for the signal $A_0(2\tau_{DQ})$ and reversible in time for the signal $A_1(2\tau_{DQ})$. This property allows us to rewrite the relation (44) for the rigid lattice as follows:



$$I_{nDQ}^{rig}\left(\tau_{DQ}\right) = \frac{1}{2}\left(1 - \frac{1}{N_s}\sum_i\left\langle\prod_j\cos\left(\tilde{\varphi}_{ij}^{ex} + \tilde{\varphi}_{ij}^{rec}\right)\right\rangle_{eq}\right). \tag{52}$$

Using expression (42) we can rewrite the expression (52) as follows:

$$I_{nDQ}^{rig}\left(\tau_{DQ}\right) = \frac{1}{2}\left(1 - \frac{1}{N_s}\sum_i\left\langle\prod_j\cos\left(\left(\int_0^{2\tau_{DQ}}\tilde{\omega}_{ij}^{(0)}(t_1)\,dt_1\right)\right)\right\rangle_{eq}\right). \tag{53}$$

Let us now consider the following quantity:

$$\Pi_i\left(2\tau_{DQ}\right) \equiv \left\langle\prod_j\cos\left(\left(\int_0^{2\tau_{DQ}}\tilde{\omega}_{ij}^{(0)}(t_1)\,dt_1\right)\right)\right\rangle_{eq}. \tag{54}$$

Decomposing the right part of it into a Taylor expansion, we obtain:

$$\Pi_i\left(2\tau_{DQ}\right) = \prod_j\left(1 - \frac{1}{2!}\int_0^{2\tau_{DQ}}dt_2\int_0^{2\tau_{DQ}}dt_1\left\langle\tilde{\omega}_{ij}^{(0)}(t_2)\tilde{\omega}_{ij}^{(0)}(t_1)\right\rangle_{eq} + \dots\right). \tag{55}$$

Considering the quantities $\tilde{\omega}_{ij}^{(0)}(t_1)$ as stochastic stationary random processes whose correlation functions have symmetry with respect to time reversal, we obtain:

$$\Pi_i\left(2\tau_{DQ}\right) = \prod_j\left(1 - \frac{1}{2!}\int_0^{2\tau_{DQ}}dt_2\int_0^{2\tau_{DQ}}dt_1\left\langle\tilde{\omega}_{ij}^{(0)}(t_2 - t_1)\tilde{\omega}_{ij}^{(0)}(0)\right\rangle_{eq} + \dots\right)$$
$$= \prod_j\left(1 - \int_0^{2\tau_{DQ}}d\tau\left(2\tau_{DQ} - \tau\right)\left\langle\tilde{P}_{ij}^{0,fl}\left(\tau;0\right)\tilde{P}_{ij}^{0,fl}\left(0;0\right)\omega_{ij}^{(0)}(\tau)\omega_{ij}^{(0)}(0)\right\rangle_{eq} + \dots\right) \tag{56}$$

Then we can make mean-field like approximation to the right part of the expression (57):

$$\Pi_i\left(2\tau_{DQ}\right) = \prod_j\left(1 - \int_0^{2\tau_{DQ}}d\tau\left(2\tau_{DQ} - \tau\right)\left\langle\tilde{P}_{ij}^{0,fl}\left(\tau;0\right)\tilde{P}_{ij}^{0,fl}\left(0;0\right)\omega_{ij}^{(0)}(\tau)\omega_{ij}^{(0)}(0)\right\rangle_{eq} + \dots\right)$$
$$\simeq \prod_j\left(1 - \int_0^{2\tau_{DQ}}d\tau\left(2\tau_{DQ} - \tau\right)\left\langle\tilde{P}_{ij}^{0,fl}\left(\tau;0\right)\tilde{P}_{ij}^{0,fl}\left(0;0\right)\right\rangle_{eq}\left\langle\omega_{ij}^{(0)}(\tau)\omega_{ij}^{(0)}(0)\right\rangle_{eq} + \dots\right) \tag{57}$$

By definition, see expression (32), we have $\tilde{P}_{ij}^{0,fl}\left(0;0\right) = 1$, therefore we can rewrite the expression (57) as the following:

$$\Pi_i\left(2\tau_{DQ}\right) \simeq \prod_j\left(1 - \int_0^{2\tau_{DQ}}d\tau\left(2\tau_{DQ} - \tau\right)\left\langle\tilde{P}_{ij}^{0,fl}\left(\tau;0\right)\right\rangle_{eq}\left\langle\omega_{ij}^{(0)}(\tau)\omega_{ij}^{(0)}(0)\right\rangle_{eq} + \dots\right)$$
$$= \prod_j\left(1 - \int_0^{2\tau_{DQ}}d\tau\left(2\tau_{DQ} - \tau\right)P_{ij}^{0,fl}\left(\tau;0\right)\left\langle\omega_{ij}^{(0)}(\tau)\omega_{ij}^{(0)}(0)\right\rangle_{eq} + \dots\right) \tag{58}$$



Remembering that we started with the approximation of the product of cosines, see the relation (54) and remembering that for a rigid lattice $\omega_{ij}^{(0)}(t) = \omega_{ij}$, it is natural to use the following approximation for the original relation (53):

$$I_{nDQ}^{rig}(\tau_{DQ}) = \frac{1}{2}\left(1 - \frac{1}{N_s}\sum_i\prod_j\cos\left(\sqrt{2\omega_{ij}^2\int_0^{2\tau_{DQ}}d\tau\left(2\tau_{DQ}-\tau\right)P_{ij}^{0,fl}(\tau;0)}\right)\right).$$ (59)

Using the approximation (46) for the value $P_{ij}^{0,fl}(\tau;0)$, after a number of calculations the expression (59) is transformed to the form:

$$I_{nDQ}^{rig}(\tau_{DQ}) = \frac{1}{2}\left(1 - \frac{1}{N_s}\sum_i\prod_j\cos\left(\omega_{ij}T_{ij}\right)\right),$$ (60)

with

$$\Omega_{ij}^2 \equiv \frac{I(I+1)}{6}\left(\sum_k{}'\left(\omega_{ik}^2 + \omega_{jk}^2\right)\right)$$
$$T_{ij} = \sqrt{\frac{2\sqrt{\pi}\tau_{DQ}}{\Omega_{ij}}erf\left(2\Omega_{ij}\tau_{DQ}\right) - \frac{1}{\Omega_{ij}^2}\left(1 - \exp\left(-4\Omega_{ij}^2\tau_{DQ}^2\right)\right)}$$ (61)

The expression (60) has the following asymptotic values:

$$I_{nDQ}^{rig}(\tau_{DQ}) = \begin{cases} \dfrac{1}{N_s}\sum_{i,j}\omega_{ij}^2\tau_{DQ}^2 & for \quad \tau_{DQ} \ll \Omega_{ij}^{-1} \propto T_2^{eff} \\[4ex] \dfrac{1}{2}\left(1 - \dfrac{1}{N_s}\sum_i\prod_j\cos\left(\omega_{ij}\sqrt{\dfrac{2\sqrt{\pi}\tau_{DQ}}{\Omega_{ij}}}\right)\right) & for \quad \tau_{DQ} \gg \Omega_{ij}^{-1} \propto T_2^{eff} \end{cases}.$$ (62)

For cases in which the spatial displacements of spins during DQ experiments cannot be neglected, the Anderson-Weiss approximation is usually used. With respect to our expression (44), it leads to the following formula:

$$I_{nDQ}(\tau_{DQ}) = \frac{1}{2}\left(1 - \frac{\sum_i\exp\left\{-\frac{1}{2}\sum_j\left\langle\left(\tilde{\varphi}_{ij}^{(0),ex} + \tilde{\varphi}_{ij}^{(0),rec}\right)^2\right\rangle_{eq}\right\}}{\sum_i\exp\left\{-\frac{1}{2}\sum_j\left\langle\left(\tilde{\varphi}_{ij}^{(1),ex} - \tilde{\varphi}_{ij}^{(1),rec}\right)^2\right\rangle_{eq}\right\}}\right).$$ (63)

If all spins have an equivalent environment, then the expression is simplified:

$$I_{nDQ}(\tau_{DQ}) = \frac{1}{2}\left(1 - \exp\left\{-\sum_j\left(\left\langle\tilde{\varphi}_{ij}^{(0),ex}\tilde{\varphi}_{ij}^{(0),rec}\right\rangle_{eq} + \left\langle\tilde{\varphi}_{ij}^{(1),ex}\tilde{\varphi}_{ij}^{(1),rec}\right\rangle_{eq}\right)\right\}\right).$$ (64)

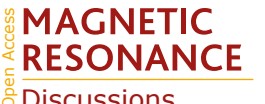

### 3.3 Estimation of characteristic flip-flop transition times in DQ experiments.

As we noted at the beginning of this section, the new expressions for the DQ signals (41) differ significantly from the simplified one (27) only at sufficiently large times $t \geq \tau_{DQ}^{fl}$, where $\tau_{DQ}^{fl}$ is the characteristic time of the flip-flop processes determined by the value $\tilde{P}_{ij}^{n,fl}\{t;0\}$, see expression (38). Now we will try to express this time through the effective spin-spin relaxation time $T_2^{eff}$, defined by the relation $g^{FID}\left(T_2^{eff}\right) = e^{-1}$, where $g^{FID}(t)$ is the FID or Hahn echo signal governed by dipolar dephasing and an approximately Gaussian initial decay. To do this, let us begin by analyzing the modified Anderson-Weiss approximation for $g^{FID}(t)$ obtained in ref. (Fatkullin et al., 2012) and written in terms of the variables of this paper:

$$g^{FID}(t) = \exp\left\{-\frac{3}{4}I(I+1)\frac{4}{N_s}\sum_{k,l}\int_0^t d\tau(t-\tau)\left\langle\omega_{kl}(\tau)\omega_{kl}(0)\right\rangle_{eq}P_{kl}^{FID,fl}(\tau)\right\}. \tag{65}$$

Since we are interested in relatively short times in the relation (64) we can put $P_{kl}^{FID,fl}(\tau) = 1$, which, combined with the assumption that all spins have the same environment, allows us to convert it to the form:

$$g^{FID}(t) = \exp\left\{-\frac{3}{2}I(I+1)\sum_l{}'\left(\left\langle\left(\varphi_{kl}^{(0)}(t;0)\right)^2\right\rangle_{eq}\right)\right\}. \tag{66}$$

If we apply the Anderson-Weiss approximation to expression (45), we obtain

$$
\begin{aligned}
P_{ij}^{n,fl}\{t;0\} &= \exp\left\{-\frac{1}{6}I(I+1)\sum_k{}'\left(\left\langle\left(\varphi_{ik}^{(n)}(t;0)\right)^2\right\rangle_{eq} + \left\langle\left(\varphi_{jk}^{(n)}(t;0)\right)^2\right\rangle_{eq}\right)\right\} \\
&= \exp\left\{-\frac{1}{3}I(I+1)\sum_k{}'\left(\left\langle\left(\varphi_{ik}^{(n)}(t;0)\right)^2\right\rangle_{eq}\right)\right\}
\end{aligned}, \tag{67}
$$

where we put, that $\sum_k{}'\left\langle\left(\varphi_{ik}^{(n)}(t;0)\right)^2\right\rangle_{eq} = \sum_k{}'\left\langle\left(\varphi_{jk}^{(n)}(t;0)\right)^2\right\rangle_{eq}$. Note that by virtue of the last approximation the values $P_{ij}^{0,fl}\{t;0\}$ and $P_k^{FID,fl}(t)$ are equal for n=0.

Let us now consider the case of the DQ experiment, n=0, i.e., no time reversal operation with respect to spin variables is performed at time $t = \tau_{DQ}$. We see that the expressions for $g^{FID}(t)$ and $P_{ij}^{0,fl}\{t;0\}$ are similar. Now we can determine the corresponding characteristic times by means of the relations:

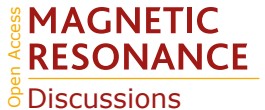

$$g^{FID}(t) = \exp\left\{-\left(\frac{t}{T_2^{eff}}\right)^{\alpha}\right\}$$

$$P_{ij}^{0,fl}\{t;0\} = \exp\left\{-\left(\frac{t}{T_{ij}^{DQ,fl}}\right)^{\alpha}\right\} ,$$

(68)

where $1 \le \alpha \le 2$ is the system dependent exponent. In solids $\alpha = 2$, in high molecular polymer melts, $1.25 \le \alpha \le 1.75$, (see, for example Kimmich and Fatkullin, 2017; Rössler et al., 2013; Fatkullin et al., 2015), for low molecular liquids $\alpha = 1$ ( see, for example Mehring, 1983; Abragam 1961). From the relations (66), (67) and (68) follows the following relation between the discussed characteristic times:

$$T_{ij}^{DQ,fl} = \left(\frac{9}{2}\right)^{1/\alpha} T_2^{eff} .$$

(69)

We see that the numerical multiplier linking the two characteristic times is quite large, even with the largest possible value

of the exponent $\alpha = 2$ : $T_{ij}^{DQ,fl} = \left(\frac{9}{2}\right)^{1/2} T_2^{eff} \approx 2.12 T_2^{eff}$ .

(70)

The latter makes it natural to consider time $\left(\frac{9}{2}\right)^{1/2} T_2^{eff} \approx 2.12 T_2^{eff}$ as a lower bound, starting from which the influence of

flip-flop processes becomes dominant. The duration of the DQ experiment is equal to $2\tau_{DQ}$, so the influence of

intermolecular flip-flop processes on the experimentally measured signal can be neglected at times

$\tau_{DQ} < \left(\frac{9}{8}\right)^{1/2} T_2^{eff} \approx 1.06 T_2^{eff}$ .

### 3.4 Numerical results and comparison with spin dynamics simulations

We now turn to comparing the results of Section 3.2, specifically the quasi-static approximation, to results of spin dynamics simulations based upon solving the Liouville-von-Neumann equation in small time steps for finite few-spin systems for the explicit BP pulse sequence (as well as simple FIDs after a 90° pulse), always assuming δ pulses. For time efficiency, we did not simulate the two different BP experiments with variable reconversion phase, but using a fixed 90° phase shift and filtering the density matrix for DQ coherences after the excitation block, thus calculating directly the DQ build-up curve. We used an earlier home-written code ( Saalwächter and Fischbach 2002) that is not optimized (no sparse-matrix algebra is implemented), which means that simulations are limited to 8 spins due to the large dimension (up to $2^8$) of the density matrix and the operators/propagators. We implemented the analytical solution, expressions (60,61), on the basis of the very same



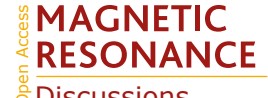

code, using the same core routines handling the dipolar interaction tensors and the same input files with spin system parameters.

As to spin systems, we aimed at mimicking a simple main-chain protonated polymer, where the chain motions provide a

fast-limit average of all conformations between two crosslinks or entanglements. This provides uniaxial averaging of all intra-chain dipolar tensors, resulting in residual coupling tensors that are all colinear (being parallel to the end-to-end distance of the chain) and reduced in magnitude by a factor of about 100 compared to the static limit (Saalwächter , 2007). A physically realistic model would have to be based on a trajectory of a molecular dynamics simulation. For simplicity, we chose to simulate cut-outs of all-trans alkane structures $(CH_2)_n$ using canonical CH and CC distances of 0.109 and 0.154 nm,

respectively, assuming tetrahedral symmetry, see the inset of Figure 1a. This model provides $r_{HH} = 0.178$ nm and thus an intra-$CH_2$ static-limit coupling constant of $D^{HH}/2\pi = 24$ Hz, see equation (2). We always detect (or calculate for) the central protons. Uniaxial averaging is implemented by symmetric three-site jumps mimicking fast rigid-body rotation (leading to a scaling of the HH dipolar couplings by -0.5 when the HH bond is perpendicular to the rotation axis), providing a situation with all-colinear dipolar tensors. To reach residual couplings corresponding to those of polymer melts, a scaling factor of

0.01 was applied to all couplings, leading to a dominant intra-$CH_2$ residual dipolar coupling constant $D_{res}^{HH}/2\pi = 122$ Hz. Another set of simulations considered a propyl fragment in g+g+ conformation (locating two outer protons in the CCC plane) with up to two additional protons located at van-der-Waals distance above either of the two central protons (with a remote coupling of $D_{HH}/2\pi \approx 9$ kHz). Powder averaging was performed over only 40 angles of β between the main axis of the averaged tensor and the magnetic field (as we simulate in the time domain, convergence was reached within the

discussed limited time intervals).

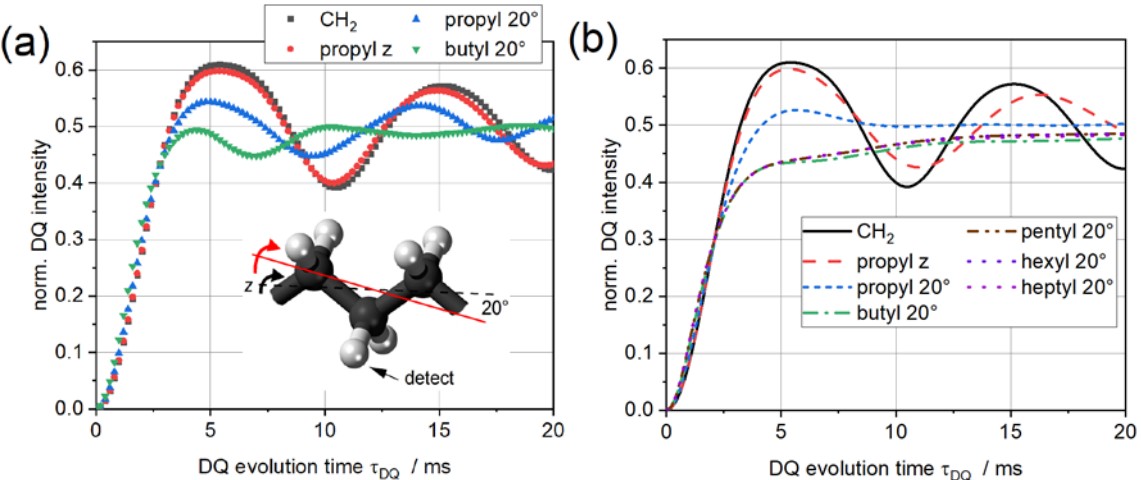

**Figure 1.** $^1$H DQ build-up curves of all-trans alkyl cut-outs rotating about the molecular long axis (*z*) or about an axis inclined by 20° (a) from spin-dynamics simulations and (b) analytically calculated from expressions (60,61).






Simulation results of DQ build-up curves are compared in Figure 1a for a $CH_2$ group (for which simulation and analytical prediction are identical) and for rotating alkyl cut-outs starting with propyl (6 protons). For rotation around the all-trans ($z$)

axis, the secondary couplings of the central $CH_2$ protons to the ones on the side are very small after uniaxial $z$-averaging, due to angles between the HH vectors and the $z$ axis being close to the magic angle. This is obvious from the very small difference of the $CH_2$ and the "propyl $z$" responses. To mimic a more complex spin system with a larger spread of couplings, we inclined the rotation axis by 20°, rending the couplings of the central $CH_2$ group to the ones on the different sides different. As a result, the coherent oscillations are significantly damped. Adding more $CH_2$ groups (with butyl being the

largest feasible spin system for the simulations) damps the oscillations even more, leading to a build-up curve that reaches the expected plateau at $I_{DQ} = 0.5$ from below.

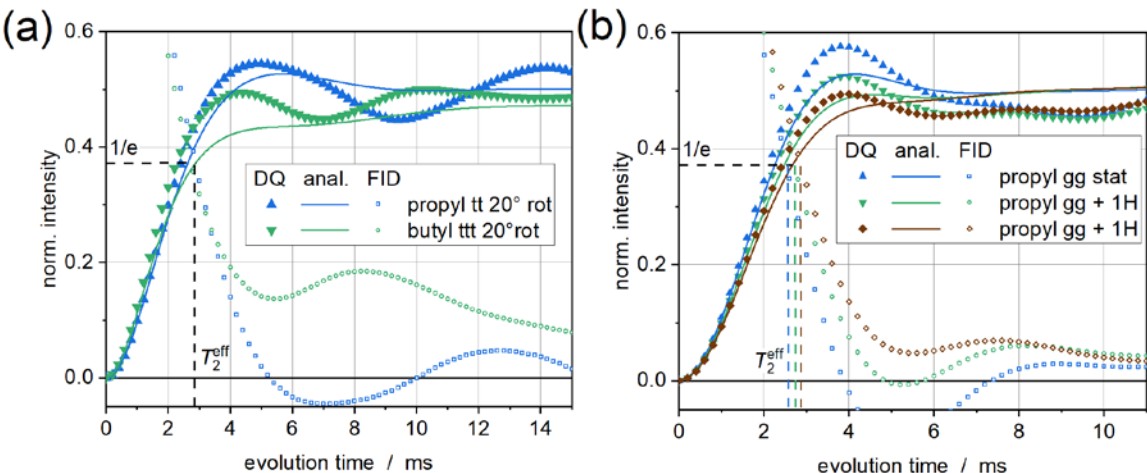

**Figure 2.** [1]H DQ build-up curves of alkyl cut-outs, comparing spin dynamics simulation results and analytical calculations, (a) in all-trans conformation rotating about an axis inclined by 20° (all couplings scaled by 0.01, see Figure 1) and (b) propyl (6 spins) in static $g^+g^+$ conformation (all couplings scaled by 0.005), with up to two additional remote protons located at van-der-Waals distance above either of the two central $CH_2$ protons ($D_{HH}/2\pi \approx 9$ kHz). Simulated FID signals are also shown to indicate $T_2^{eff}$.


The analytical results shown in Figure 1b, for the first time possible for systems beyond a spin pair, mimic these trends surprisingly well. Notably, the changes in the build-up curves upon adding more $CH_2$ groups, which is easily possible in the analytical calculations, does not change the result significantly. For a more quantitative comparison, we directly compare in Figure 2a the simulations and analytical calculations for the propyl 20° and butyl 20° cases.



While the agreement between the former pair is very satisfactory, large deviations are observed in the latter case. This may well be due to the finite spin system and specificities related to the all-colinear dipolar tensors. In Figure 2 we also plot simulated FIDs, which can be used to extract the $T_2^{\text{eff}}$. Up to $\tau_{DQ} = T_2^{eff}$, simulated and analytical results match within 15%.

To explore the effect of the all-tensor colinearity in these calculations, we add in Figure 2b simulations and calculations for a static 3D spin system (with a dipolar scaling factor of 0.005 to arrive at a similar coupling magnitude as before), in this case

of propyl fragment in g+g+ conformation, optionally adding two remote protons. It is observed that the agreement between simulations and analytical solutions within $T_2^{\text{eff}}$ is generally even better, confirming the hypothesis that a three-dimensional distribution of variable coupling tensors is maybe a better basis for the application of the AW approximation inherent to expression (67).

Thus, for cases when we have a large scatter in the coupling constants for different spins, this result, in our opinion, can be considered as quite satisfactory. The improvement of the result requires a more detailed treatment of flip-flop processes between different spins than we have done in the transition from the relation (45) to the relation (46), which does not take into account the returns of spin polarization during spin diffusion to the initial spin, which will lead to a slower decay of the function (46) at times $\tau_{DQ} \geq T_2^{eff}$. Also at the discussed times it becomes necessary to improve the approximation (32) due to

the simultaneous exchange of two different pairs of spins by their mutual spin polarizations, see the remark after formula (37). It may also be important to further develop the ideas presented in refs ( Bochkin et al., 2022; Bochkin et al., 2024; Fel'dman et al., 2022) . Besides, a more detailed assessment, using more realistic and much larger spin systems, requiring dedicated simulation software, was beyond the scope of the present work.

**4    Conclusions.**

The mathematical identity (7) allows us to reformulate the derivation of experimentally measured signals in DQ experiments in such a way that taking into account the effects of inter-spin flip-flop processes is natural and simple.  In this way, it was possible for the first time to derive an analytical calculation of DQ build-up curves in multi-spin systems. From a formal

point of view, it all comes down to redefining the phases of mutual rotations of spins induced by the DQ Hamiltonian (4), compare the relations (27) and (41), (44). The influence of flip-flops leads to the fact that the phases turn out to be linearly dependent on the conditional probabilities $\tilde{P}_{ij}^{n,fl}(t)$  that the corresponding pair of spins did not participate in flip-flops with any other spin of the system during the time interval t, see expressions (40) and (42). The structure of the DQ Hamiltonian (4) itself is such that the probabilities of the flip-flop processes induced by it are 2 times smaller than those induced by the

secular part of the Hamiltonian of magnetic dipole-dipole interactions (1). The latter allows us to neglect the effects of flip-flop processes in DQ experiments and use the simplified description given by the relations (27) on sufficiently long time

**MAGNETIC RESONANCE**
Open Access Discussions

intervals in units of effective spin-spin relaxation time $t < 2.12 T_2^{eff}$ . A comparison of the predictions with spin dynamics simulations of simple, small spin systems of different sizes provided a promising, near-quantitative agreement for $\tau_{DQ} \leq 1.06 T_2^{eff}$ , yet the origin of existing deviations for longer times requires further work.


**Acknowledgments:**

The authors gratefully acknowledge financial support by the Deutsche Forschungsgemeinschaft (DFG) in the framework of the SFB-TRR 102 (project ID 824189853844).

**Competing interests**:

Kay Saalwächter, one of the co-authors of the article, is a member of the editorial board of MR.

**Data availability :**

The simulation codes as well as datasets generated and analyzed for this study as they appear in the figures of this article can
be found in the Zenodo repository: https://doi.org/10.5281/zenodo.13628349 . Note for reviewers: this link will only be activated upon publication


**Author contributions:**

**N. F. Fatkullin**: Conceptualization (equal); Investigation/analytical theory ; Supervision ; Writing – Original Draft, Review and Editing (equal).

**I. V. Brekotkin**: Validation ; Investigation (equal);

**K. Saalwächter**: Conceptualization (equal); Investigation/simulations ; Funding Acquisition ;  Writing – Review & Editing (equal).

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
