# Peer review of "Analytical treatment of proton double-quantum NMR intensity buildup: multi-spin couplings and the flip-flop term"

_Magnetic Resonance, 2024_

## Author Comment (AC4)

Response to anonymous Referee #2

*We are grateful to the second reviewer for his careful review of the manuscript and comments, which will incorporate in the new version of our paper.*

The manuscript describes an analytical treatment of DQ excitation in solids in large spin systems where dynamics and spin flip-flop processes lead to time modulations of the dipolar couplings. In an iterative process of interaction-frame transformations, it is shown that the DQ build up can be calculated analytically while taking the action of the dynamics and spin flip-flop transitions into account. For this an additional interaction-frame transformation with the part of the DQ Hamiltonian that commutes with I^z (flip-flop part) is performed. This leaves just a yy part of the Hamiltonian that commutes with itself at all times and allows the calculation of an analytical phase factor that describes the DQ intensity. Results are compared to numerical simulations in relatively small spin systems (6 spins) which shows reasonable agreement with the analytical calculations. Can the authors at least comment whether they expect better agreement if state of the art spin system sizes of 10-12 spins (doable without sparse matrix implementations) would give better agreement?

*To discuss possible improvements in the agreement between simulations and predictions, one should consider that our (still limited) approximation is for now limited to times $\tau_{DQ} \leq T_2^{eff}$ (see lines 470-480); for longer times, additional work (e.g. evaluating higher-order corrections) is required. Generally, of course, we expect better agreement between simulations and either better predictions or experimental results for more spins. Our (outdated) code is unfortunately limited to 9 spins at present. But the main limitation is in fact the consideration of a realistic spin system, where our alkyl cut-outs are already a bit of a stretch. With nowadays' improved simulation codes such as SPINACH, handling tens of spins, we shall in the future focus on trajectories from MD simulations for the most realistic modelling of fast local conformational dynamics and in particular inter-molecular couplings. We will add corresponding remarks in the paper.*

The manuscript is clearly inside the scope of MR and presents interesting new results that are of interest to the MR readership. However, I believe that the presentation could be improved and that there are a larger number of typos and very colloquial expressions that make it difficult to follow the text.
The authors start from a secular high-field truncated dipolar Hamiltonian (Eq (1)) which has the usual form. But then the authors say: "The Hamiltonian (1) in this paper plays the role of the original spin-lattice interaction Hamiltonian. In the lowest order of perturbation theory, it induces in a spin system 0-quantum transitions (dephasing processes or 0-quantum coherence) by terms proportional to .... 1-quantum transitions (transitions with a flip of one of a pair of interacting spins or 1-quantum coherence) and 2-quantum transitions (coordinated transitions of two spins or 2-quntum coherence) by terms proportional ...." (equation cut out of text). Why would this (static) ZQ Hamiltonian induce DQ or SQ transitions. This would be true for the untruncated dipolar Hamiltonian but not for the high-field truncated one used here, or am I missing something?

*The reviewer is right: the properties of the full Hamiltonian of magnetic dipole-dipole interactions, which was written in this part of the paper in the very first version, are attributed to its secular part. Corresponding changes will be made in the revised version of the paper:*
*" The Hamiltonian (1) in this paper plays the role of the original spin-lattice interaction Hamiltonian. In the lowest order of perturbation theory, it induces in a spin system only 0-quantum transitions (dephasing processes or 0-quantum coherence) through terms proportional to $\hat{I}_i^z \hat{I}_j^z$ and flip-flop processes proportional to $\hat{I}_i^+ \hat{I}_j^- + \hat{I}_i^- \hat{I}_j^+$. We also note that the non-secular part of the Hamiltonian of magnetic dipole-dipole interactions inducing the processes of spin-lattice relaxation describes the*

*processes of 1-quantum and 2-quantum coherence. However, at the times of interest $t \ll T_1$   they*
*can be neglected."*

There are a lot of very colloquial and not very clear and precise statements that could be improved. Examples are:

Line 74: "In solids, the joint effect of mentioned factors, in the limit $\Delta \to 0$ , where $\Delta$ is time interval between the nearest RF pulses" which joint effects are meant here?

*This will be improved in the following way:*
*" In solids, the joint effect of the BP sequence and the Hamiltonian (1) , in the limit $\Delta \to 0$ , where $\Delta$*
*is time interval between the nearest RF pulses, the spin system's time evolution can be described*
*in terms of an effective DQ Hamiltonian having the following structure:*

$$\hat{H}_{DQ}^{n} = \left(-1\right)^{n\theta\left(t-\tau_{DQ}\right)} \sum_{i<j} \hbar\omega_{ij} \left(\hat{I}_i^y \hat{I}_j^y - \hat{I}_i^x \hat{I}_j^x\right) = \left(-1\right)^{n\theta\left(t-\tau_{DQ}\right)} \sum_{i<j} \frac{\hbar\omega_{ij}}{2}\left(\hat{I}_i^+ \hat{I}_j^+ + \hat{I}_i^- \hat{I}_j^-\right), \qquad (3)$$

*where $\theta(x)$ is the Heaviside step function, $n = 0,1$. The case $n = 0$ corresponds to the first*
*mentioned version of DQ experiment without phase change, i.e, to the signal $A_0\left(2\tau_{DQ}\right)$, and $n = 1$ to*
*the second version with 90° phase shift, i.e. to the signal $A_1\left(2\tau_{DQ}\right)$. "*

Line 78: "Note, that n = 0 corresponds to the first mentioned version of DQ experiment no phase change and n = 1 to the second version 90° phase shift." The English is wrong (replace "no" with "without") and it could be stated clearly that you refer to A-0 and A_1 signals here.

*This will be improved.*

Line 84: "At time moment DQ t =t the operator (3) changes time for the case, when n = 1." A Hamiltonian cannot change time. I can change the sign.

*This is considered in the above passage referring to eq. (3).*

Line 95: "where the role of the zero Hamiltonian is sum of the lattice Hamiltonian and the Hamiltonian of Zeeman interaction of investigated spins with an external magnetic field." What is a zero Hamiltonian? It is defined later but I find this a very colloquial expression. Maybe the "dominant part of the Hamiltonian" would be a better term.

*To be improved:*
*"…zeroth-order Hamiltonian, i.e. the zero-point Hamiltonian or the dominant part of the Hamiltonian*
*(see also expr. (8)), is sum of the lattice Hamiltonian and the Zeeman Hamiltonian of the*
*investigated spins with an external magnetic field..."*

There are more such unclear statements and the manuscript would profit from a careful reading and checking for consistency of notation (see also minor points) as well as the English language.

*The revised version will be carefully edited.*

I think the manuscript would benefit from making section 3 really a discussion without pages and pages of equations that could be moved to an appendix. At the moment section 3.1-3.3 is more a theory section that is very hard to follow and gives little insight for the non-specialist.

*We will move these sections of the paper to the theoretical part (as we believe that moving it into an Appendix would disrupt the flow of arguments unnecessarily)*

Minor points - many typos (?) or unclear expressions in equations:

line 39: two full stops

line 58: quantum instead of quntum

Eq. (4): This should probably be a \omega_{ij}(t) and not \omega_{ij}^{(n)}(t) which is only defined in (5)

*All the above corrections will be implemented*

Line 102: Give a reference for AW approximation.

*We will add references to:*
*Anderson P.W., P.R. Weiss P.R.: Exchange narrowing in paramagnetic resonance, Rev. Mod. Phys., 25, 1, pp. 269-276, 1953.*
*Kimmich R.: NMR Tomography Diffusometry Relaxometry, Springer-Verlag Berlin Hedelberg (1997)*

line 164: Are I_z and I^z the same or different quantities?

*They are the same, to be improved.*

line 164: What is H_0 here? the Zeeman Hamiltonian or the flip-flop part of the DQ Hamiltonian?

*Here, the zeroth-order average Hamiltonian is still given by expression (12): Zeeman + lattice. This will be clarified.*

Eq. 14 and 15: Is U_0(t) on the first line of Eq. 14 the same as U_0^{(DQ,n)} on the first line of Eq. 15?

*This misprint will be fixed.*

Eq. 15: Is H^{(0)}_{DQ} the same as H^{(n)}_{DQ} in Eq. 16

*Yes, to be stated clearly.*

Line 205: There is no section 2b

*To be fixed.*

Eq. 32: Is {\tilde{I}_i^y\tilde{I}_j^y}^{fl}_t the same as the (I_i^yI_j^y)^{fl}_t on line 205 and Eq. 31? If not then please define.

*Yes, to be improved.*

There are probably many more that I did not see.

*We will do our best to carefully check the manuscript again.*